# Nanosensor Based on Thermal Gradient and Machine Learning for the Detection of Methanol Adulteration in Alcoholic Beverages and Methanol Poisoning

**DOI:** 10.3390/s22155554

**Published:** 2022-07-25

**Authors:** Matteo Tonezzer, Nicola Bazzanella, Flavia Gasperi, Franco Biasioli

**Affiliations:** 1Research and Innovation Centre, Fondazione Edmund Mach, via E. Mach 1, 38010 San Michele all’Adige, Italy; flavia.gasperi@fmach.it (F.G.); franco.biasioli@fmach.it (F.B.); 2Center Agriculture Food Environment, University of Trento/Fondazione Edmund Mach, via E. Mach 1, 38010 San Michele all’Adige, Italy; 3IMEM-CNR, Sede di Trent o-FBK, Via alla Cascata 56/C, Povo, 38123 Trento, Italy; 4Department of Physics, Università degli Studi di Trento, Povo, 38123 Trento, Italy; nicola.bazzanella@unitn.it

**Keywords:** metal oxide, tin oxide, gas sensor, resistive sensor, nanowires, methanol, ethanol

## Abstract

Methanol, naturally present in small quantities in the distillation of alcoholic beverages, can lead to serious health problems. When it exceeds a certain concentration, it causes blindness, organ failure, and even death if not recognized in time. Analytical techniques such as chromatography are used to detect dangerous concentrations of methanol, which are very accurate but also expensive, cumbersome, and time-consuming. Therefore, a gas sensor that is inexpensive and portable and capable of distinguishing methanol from ethanol would be very useful. Here, we present a resistive gas sensor, based on tin oxide nanowires, that works in a thermal gradient. By combining responses at various temperatures and using machine learning algorithms (PCA, SVM, LDA), the device can distinguish methanol from ethanol in a wide range of concentrations (1–100 ppm) in both dry air and under different humidity conditions (25–75% RH). The proposed sensor, which is small and inexpensive, demonstrates the ability to distinguish methanol from ethanol at different concentrations and could be developed both to detect the adulteration of alcoholic beverages and to quickly recognize methanol poisoning.

## 1. Introduction

Methanol, alcohol produced naturally (in minimal quantities) in the distillation and production of alcoholic and even non-alcoholic beverages, can be highly toxic to human health. Methanol poisoning, which usually occurs by ingestion, can lead to irreversible tissue damage, especially to the eyes and nervous system, or even death [1]. This happens because methanol is metabolized by the body to form formic acid, formate, and formaldehyde [2], which are very toxic [3]. Outbreaks of methanol poisoning occur frequently in many countries, with hundreds of deaths due to adulterated alcohol [4]. Examples include the 959 cases in Iran (October 2018) [5], 237 in Cambodia (May 2018) [6], 45 deaths in Malaysia (October 2018) [7], and more than 250 deaths in India (February 2019) [8,9]. Furthermore, methanol is often used as a solvent or chemical raw material in chemical laboratories and plants [10], which creates a risk of intoxication even by inhalation or absorption from the skin [11].

Methanol intoxication is usually detected in the blood by analytical techniques such as gas-liquid chromatography and blood gas analysis [12], which require qualified personnel and are costly in terms of both time and money. For these reasons, they are not readily available, especially in non-urban areas and in developing countries, where outbreaks are more frequent [13]. Levels of methanol intoxication can also be determined non-invasively in exhaled breath [14], such as what is performed daily by law enforcement with ethanol [15]. Although the average concentration of methanol in the breath of healthy people is less than 1 ppm, concentrations up to 10 ppm may be found in some cases [14], while concentrations above 150 ppm are considered a symptom of severe intoxication [1]. The recommended airborne exposure limit (REL) by the American National Institute for Occupational Safety (NIOSH), Occupational Safety and Health Administration (OSHA), and American Conference of Governmental Industrial Hygienists (ACGIH) is 200 ppm averaged over a 10-h work shift [16].

The challenge is therefore to distinguish methanol from ethanol and quantitatively estimate their amount with a small, inexpensive, and portable device. Similarly, such a device would also be important for screening alcoholic beverages in order to prevent methanol poisoning.

Gas chemosensors are ideal candidates for this application, as they are simple to make and use, inexpensive, and miniaturizable [17]. Chemoresistive sensors based on metal oxide nanostructures have been shown to detect various analytes at concentrations below parts per million (ppm) in a very short time [18]. Unfortunately, these materials are not very selective even if their morphology and structure are optimized to increase porosity and reactivity [19]. Due to this poor selectivity, it is difficult to distinguish two similar molecules such as ethanol and methanol. Although SnO_2_ is one of the metal oxides with the best properties and therefore most used for gas sensors, even the finest nanostructures such as cross-linked porous nanosheets [20] and hollow nanoparticles [21] exhibit very similar responses to these two gases, making them difficult to distinguish. Therefore, chemoresistors are usually joined in arrays called electronic noses, which exploit different materials to obtain good selectivity [22,23]. The interest in electronic noses leads to the study of the most innovative materials, such as graphene and graphene oxide, and the use of algorithms such as the support vector machine (SVM) allows one to obtain good quantitative results [24]. Most resistive electronic noses use different metal oxides (SnO_2_, ZnO, WO_3_) and different surface decorations with metal nanoparticles (Ag, Pt, Pd) to obtain good performance also in the detection of ethanol and methanol [25]. Being composed of different sensors based on different materials (metal oxides, polymers, small conjugated molecules, and others) that require different working conditions, heaters, and electrodes for individual signal acquisition, current electronic noses are still rather complex and expensive.

Here we describe a gas sensor, based on tin oxide (SnO_2_) nanowires, that aims at selectivity, not using different materials but rather different operating temperatures. The detection mechanism is based on the chemical reactions that take place on the surface of the nanowires, where the volatile molecules react by releasing or absorbing electrons, changing the resistance of the sensor. The material response changes with both the temperature and volatile compound concentration. This produces a “thermal/chemical fingerprint” which can be the basis of an electronic nose [26].

This approach has already been demonstrated on agrifood products by evaluating the freshness of meat and fish [27,28]. In practice, we join the responses at different temperatures (as if they came from different sensors) and combine them in multidimensional points. Analyzing them with machine learning algorithms and multivariate statistical analysis techniques (principal component analysis, support vector machine, linear discriminant analysis), we demonstrate that the sensor is not only able to distinguish ethanol and methanol, but also to estimate their concentration. Considering the aforementioned hazard and intoxication thresholds, the sensor was tested in a concentration range of 1 to 100 ppm. The sensor has been tested with different concentrations of ethanol and methanol under more realistic and difficult conditions (relative humidity of 25 to 75%) and proved to be able to distinguish the two alcohols under any conditions. This performance makes the nanowire-based sensor an excellent candidate for rapid and inexpensive screening for the presence of methanol in both intoxicated patients and potentially adulterated beverages.

## 2. Materials and Methods

### 2.1. Synthesis of SnO_2_ Nanowires

A forest of tin oxide nanowires was initially grown by chemical vapor deposition (CVD). An alumina boat filled with pure tin monoxide was placed as an evaporation source in the center of an oven (Lindberg Blue M, Thermo Fisher Scientific, Waltham, MA, USA) where the temperature is highest. Close to it (1 cm), a silicon substrate of approximately 1 × 2 cm^2^ was placed, on which a thin gold film (thickness of approximately 5 nm) acted as a catalyst. The quartz tube was then pumped down to 10^−2^ mbar and purged with high-purity argon (99.999%), repeating these two steps three times, and finally, the system was pumped down to 8 × 10^−3^ mbar. While the system was in a vacuum, the temperature was increased from room temperature (23 °C) up to 800 °C at a rate of 25 °C/min and then the oven was held at 800 °C for five minutes. At this point, an oxygen flow of 0.35 standard cubic centimeters (sccm) was injected into the system in order to start the growth of the nanowires. The growth of the nanostructures, which follows the gold-catalyzed solid liquid vapor (VLS) mechanism [29], lasted 30 min, after which the system was shut down and allowed to cool. At the end of the growth process, the sample surface showed a homogeneous white film.

### 2.2. Nanowires Characterization

The morphology of the SnO_2_ nanowires was studied by secondary electron microscopy (SEM) with a Hitachi S-4800 (Tokyo, Japan). The structure of the nanowires was investigated by X-ray diffraction (XRD) using a Philips Xpert Pro diffractometer (Malvern Panalytical, Malvern, UK) working at 40 kV with CuKα radiation.

### 2.3. Sensor Fabrication

The nanowires were transferred to another substrate via sonication and drop-coating. The sample with the nanowire forest obtained from the CVD was ultrasonicated in dimethylformamide for ten seconds to obtain a dispersion of nanowires. A few drops of this dispersion were deposited on a piece of silicon wafer with a 300 nm layer of thermally grown oxide. Two interdigitated Ti/Pt electrodes were then deposited via sputtering and UV lithography on top of the nanowires, so that they formed a chemiresistor. The device was subjected to eight-hour thermal annealing at 500 °C in nitrogen in order to stabilize the structure and electrical characteristics [18].

### 2.4. Gas Sensor Measurements

The chemoresistive sensor was placed on a heatable sample holder in a vacuum chamber connected to high-purity gas cylinders through mass flow controllers and a mixing chamber. Two microprobes were connected to the metal electrodes in order to read the resistance of the nanowires with a multimeter (Keithely 2410, Cleveland, OH, USA) controlled by data acquisition software (LabView, National Instruments, Austin, TX, USA). The sensor was tested at five different temperatures (180–300 °C) over a concentration range of ethanol and methanol ranging from 1 to 100 ppm. The sensor response was defined as S = R_air_/R_gas_, where R_air_ and R_gas_ are the resistance of the sensor in the air and in the presence of gas, respectively. The measurements on the liquid mixtures were carried out by placing the sensor approximately 1 cm above the vessel with the alcohol mixture in the measurement chamber, letting the system reach equilibrium.

### 2.5. Machine Learning Techniques

Since a resistive sensor provides a one-dimensional response (a single pure number, a ratio between two electrical values), it is inherently non-selective. For this reason, the sensor responses at five different working temperatures (180, 210, 240, 270, and 300 °C) were combined to create 5-dimensional points to be processed with multivariate statistical analysis techniques [26]. The 5D points obtained were analyzed with different techniques in order to evaluate different aspects of the sensor performance. Principal component analysis (PCA) was used to graphically visualize the relationships between gas measurements, as it reduces the dimensions (from five to two) while maintaining as much information as possible. Linear discriminant analysis (LDA) was used to quantitatively evaluate the sensor’s ability to distinguish the two gases, as it maximizes the separation between classes (in our case the two gases at different concentrations). To confirm the classification of the LDA, a Euclidean UPGMA clustering was also used. The support vector machine (SVM) was used to obtain an estimate of the gas concentration through linear regression. SVM regression measurements were performed using concentrations 1, 3, 5, 10, 20, 50, and 100 ppm as the training set and 2, 4, 8, 15, 30, and 80 ppm as the test set. The classification of the measurements with LDA in humidity and the measurements of alcoholic mixtures were carried out with cross-validation, using all the possible permutations of training and test data.

## 3. Results and Discussion

### 3.1. Nanowires Characterization

The nanowires that made up the white layer obtained from CVD were initially studied by secondary electron microscopy to study the morphology of the nanomaterial. Figure 1a confirms that the material is a forest composed of long, thin nanowires arranged chaotically. The nanowires are several microns long and have constant and rather homogeneous diameters of approximately 40–70 nm. Figure 1a is blurred because of the accumulated charge due to the very high electrical resistance of the nanowires.

The composition and structure of the nanowires were investigated by means of X-ray diffraction. The top pattern in Figure 1b shows many intense and sharp peaks, which can be easily assigned to the reflections of the SnO_2_ tetragonal phase, with lattice parameters of a = b = 4.742 and c = 3.186 Å. As can be seen in the pattern in the lower part of Figure 1b (in blue), the experimental peaks aptly match those in the reference JCPDS n. 77-0450. The pattern in Figure 1b does not show other phases besides the tetragonal SnO_2_, nor peaks due to impurities or amorphous contributions, and therefore confirms the good crystallinity of the nanowires.

### 3.2. Traditional Gas Measurements

The performance of the gas sensor based on SnO_2_ nanowires was initially studied in a traditional way. The dynamic response of the device at various temperatures is shown in Figure 2a,b. The sensor was subjected to different concentrations of ethanol (Figure 2a) and methanol (Figure 2b) ranging from 1 to 100 ppm. Both graphs show that the response increased rapidly when the gas was injected and decreased even more rapidly when the gas was replaced by dry air. The intensity of the response increases with increasing gas concentration, as expected, and the signal fully recovers when dry air is injected, with no noteworthy drifts. The graphs in Figure 2c,d show the response of the sensor as a function of the gas concentration (ethanol in Figure 2c and methanol in Figure 2d). The response clearly increases with concentration, initially more markedly and then more slowly, and increases with increasing operating temperature. The graphs in Figure 2, although different, are quite similar: The response increases with the gas concentration and temperature for both ethanol and methanol. The traditional analysis of the sensor response does not allow us to distinguish the two gases.

### 3.3. Machine Learning: Visualization and Classification

To achieve selectivity and to be able to distinguish ethanol from methanol, the responses of the sensor at the five temperatures were combined in 5D points and analyzed with different techniques of multivariate statistics and machine learning. The first technique used is principal component analysis (PCA), a technique that allows the dimensions to be reduced from five to two (so that points can be visualized) while retaining maximum information from the original data. The graph of the first two main components is shown in Figure 3, where the points relating to ethanol are green and those relating to methanol are violet. It is clear that the points relating to the two gases lie on different lines, and it is therefore easy to distinguish them qualitatively. The points of each gas lie on a line because they are measurements at different concentrations: The leftmost points are the measurements at 1 ppm, and moving to the right, the points are relative to measurements at higher concentrations, up to 100 ppm (they are the same concentrations as in Figure 2a,b). Figure 3 intuitively demonstrates that the sensor is able to distinguish the points relating to the two gases, but it is not enough. For this reason, linear discriminant analysis (LDA) was also used, which is shown in the inset of Figure 3 and confirms how the points relating to the two gases are clearly distinct, also in a quantitative way. This means that the gas sensor is able to distinguish the two gases despite the different concentrations tested.

### 3.4. Machine Learning: Quantification

To assess the danger of methanol poisoning or an adulterated drink, it is necessary not only to detect its presence, but also its concentration (in the breath or in the drink). To obtain an estimate of the gas concentration, a support vector machine (SVM) was used, a supervised technique that uses the first set of data as “calibration” [30,31]. The data used for Figure 2 was used as a training set, while other concentrations (2, 4, 8, 15, 30, and 80 ppm) were used to test the sensor performance in estimating the gas concentration. The concentrations of the test dataset were chosen halfway between those of the training dataset in order to make the work more difficult for the sensor and to obtain the estimate in the worst realistic conditions. The linear support vector machine performed a regression in the five-dimensional space and the sensor provided the estimates shown in Figure 4.

The X-axis shows the nominal concentration tested, while the estimate provided by the sensor is shown on the Y-axis. The diagonal therefore represents the perfect estimate: The closer a point is to it, the more correct the estimate is.

The blue dots in Figure 4a are related to ethanol, and are very close to the diagonal, demonstrating good sensor accuracy. In fact, the mean absolute error of the sensor on the ethanol concentration is only 3.1 ppm. The green dots in Figure 4b refer to methanol, and are also close to the diagonal, confirming an accurate estimate for methanol as well. The mean absolute error for methanol is 3.3 ppm. It can be seen in both plots that the sensor estimate is always higher than the nominal value, and this is useful since it is more dangerous to underestimate the methanol concentration than to overestimate it, whether in the case of an intoxicated person or an adulterated drink.

### 3.5. Relative Humidity

The measurements shown in the previous graphs were obtained in dry air, but both the breath and the headspace of a drink have high relative humidity. For this reason, the sensor has been tested in different relative humidity conditions, in order to understand how its performance varies. To statistically evaluate the performance of the sensor, three concentrations of each gas (1, 10, and 100 ppm) were chosen, and each measurement was repeated eight times, in order to evaluate the repeatability of the measurement. This procedure was repeated at different relative humidity values: 0, 25, 50, and 75%.

The five-dimensional space was reduced by principal components analysis, and Figure 5 shows the results in 2D plots. Figure 5a shows the points in dry air (0% relative humidity, RH), and the groups of points related to each concentration of each gas are colored differently (shades of blue for ethanol and green for methanol).

At the bottom, from left to right, there are the concentrations of ethanol (1, 10, and 100 ppm), while from the bottom left, going up diagonally, there are the concentrations of methanol, in the same order. In Figure 5b, there are the points obtained at 25% RH, in Figure 5c those at 50% RH, and in Figure 5d those at 75% RH. The arrangement of the point groups is roughly always the same.

To better compare the performance under various humidity conditions, the graphs were made with the same dimensions. The distribution of the groups of points is similar in the four graphs, but it can be noticed how, as the relative humidity increases, the groups of points tend to become closer. On the one hand, the points relating to the highest concentrations of ethanol move upwards slightly, closing the gap with those of methanol. On the other hand, the points relating to methanol at higher concentrations drop a lot towards those of ethanol and towards the lower concentrations on the left. Qualitatively, this shows that the discrimination between the various concentrations of the various gases becomes a little more difficult. Unfortunately, PCA plots are purely qualitative and influenced by the perception of the human eye. To obtain a less subjective idea of the sensor performance, linear discriminant analysis was applied, the results of which are shown in Table 1.

Only one table is shown since the results in the four relative humidity conditions are the same and indicate an accuracy of 100%. Table 1 therefore summarizes the classification of the sensor for the various measurements, demonstrating that the device is able to distinguish the gas and its concentration in all humidity conditions.

To confirm these results, a Euclidean UPGMA clustering was also applied, the results of which are shown in Appendix A. In addition, in that case, the sensor demonstrates the ability to distinguish ethanol and methanol and their concentrations.

### 3.6. Experimental Measurements in Realistic Conditions

To verify its performance under realistic conditions, the sensor was tested on mixtures of alcohol in distilled water. Since most of the spirits consumed in the world (whiskey, gin, vodka, cachaça, tequila, grappa) have an alcohol content of approximately 40°, we have simulated bottles of this type by making alcohol mixtures at 40° in distilled water.

Since 100 mL of methanol is considered the minimum lethal dose in humans [32], we made three types of samples: Ten samples contained a water/ethanol solution (40%), simulating a safe commercial distillate; in ten samples, 100 mL of ethanol was replaced with methanol; and in another ten samples, 50 mL of ethanol was replaced with methanol (to simulate a less dangerous dose).

A graph of the first two principal components is shown in Figure 6, in order to illustrate the relationships between the clusters of points relative to the different mixtures. The ellipses indicate 90% confidence intervals. The points from mixtures containing methanol are partially overlapped, while the points from mixtures with only ethanol are distinct from the others.

The partial overlap between the ellipses of the samples with 50 and 100 mL of methanol is due to the limitedness of the 2D graph. To quantitatively verify the sensor performance in realistic conditions, linear discriminant analysis was used, which showed a correct classification in 100% of cases. This means that the sensor can perfectly distinguish the three classes of alcoholic mixtures, but above all, more importantly, the safe mixtures (without methanol) from the toxic ones (with methanol).

Although more tests must certainly be performed, we believe that these results demonstrate the ability of the electronic nose based on a thermal gradient to also face applications in the field of food safety.

## 4. Conclusions

A single chemoresistive sensor based on SnO_2_ nanowires was used to distinguish methanol from ethanol and measure its concentration. The responses of the sensor at five working temperatures were combined in five-dimensional points in order to then be processed with multivariate statistical analysis and machine learning techniques. Using principal component analysis, linear discriminant analysis, and a support vector machine, the sensor was able to accurately distinguish methanol from ethanol and measure the concentration of the two gases with an average error of 3 parts per million in approximately 2–3 min. The sensor has proven the ability to distinguish the type of gas and its concentration in all conditions of relative humidity. The sensor correctly classified 100% alcoholic mixtures at 40° (simulating vodka, whiskey, gin) in real conditions. These performances demonstrate that this approach can be effective in detecting methanol-intoxicated patients or detecting the presence of methanol in alcoholic beverages in lieu of more expensive and time-consuming analytical techniques.

## Figures and Tables

**Figure 1 sensors-22-05554-f001:**
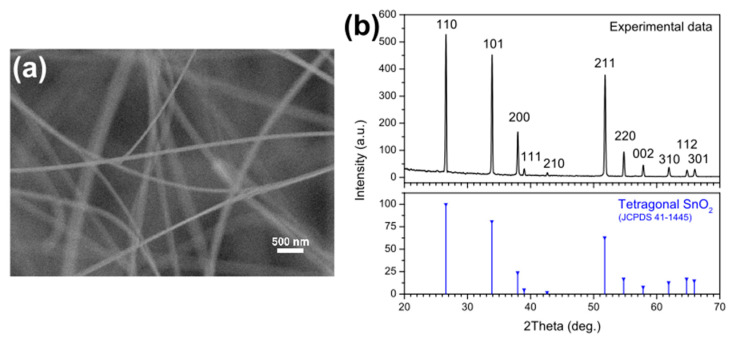
(**a**) SEM image and (**b**) XRD pattern of the SnO_2_ nanowires.

**Figure 2 sensors-22-05554-f002:**
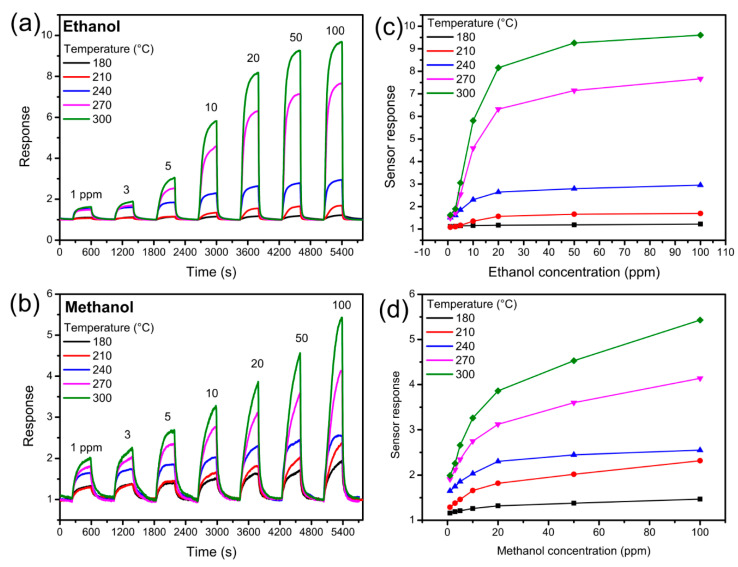
Dynamic response of the sensor to different concentrations of (**a**) ethanol and (**b**) methanol at different temperatures; response as a function of gas concentration for (**c**) ethanol and (**d**) methanol at different operating temperatures.

**Figure 3 sensors-22-05554-f003:**
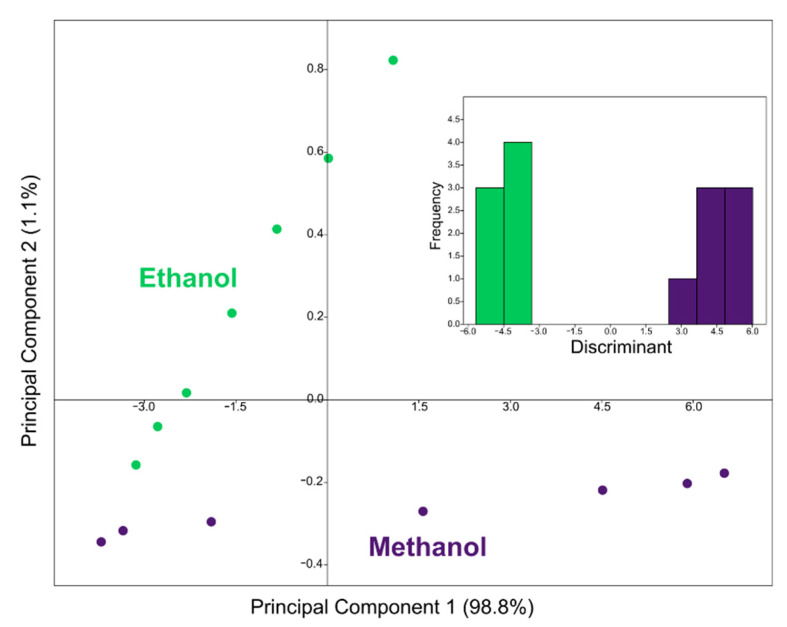
Plot of the first two main components illustrating the measurements at different concentrations (1 to 100 ppm) of ethanol (in green) and methanol (in violet). Inset: Linear discriminant analysis of the points relating to the two gases, which shows how they are separated and distinguishable.

**Figure 4 sensors-22-05554-f004:**
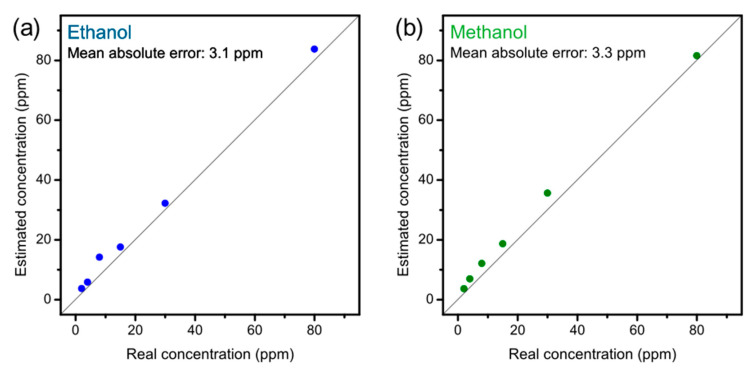
Estimation of the concentration of (**a**) ethanol and (**b**) methanol, obtained with a linear regression by means of a support vector machine.

**Figure 5 sensors-22-05554-f005:**
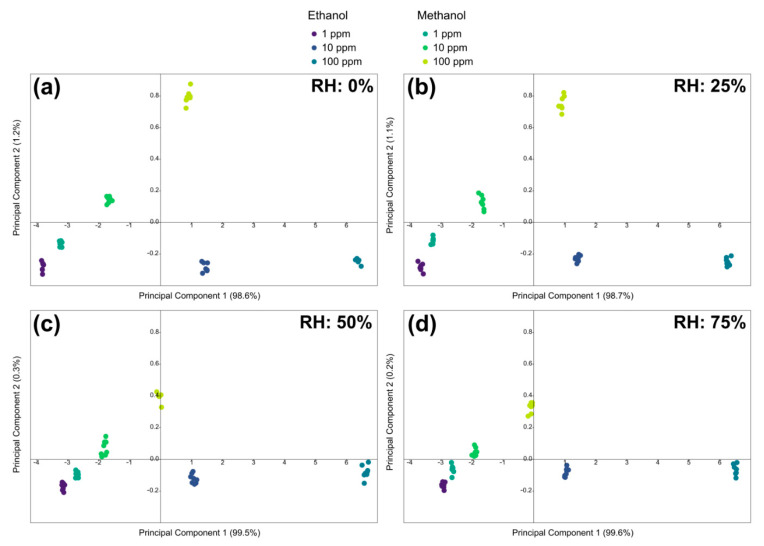
Plots of the principal components for concentrations of 1, 10, and 100 ppm of ethanol and methanol at (**a**) 0%, (**b**) 25%, (**c**) 50%, and (**d**) 75% relative humidity.

**Figure 6 sensors-22-05554-f006:**
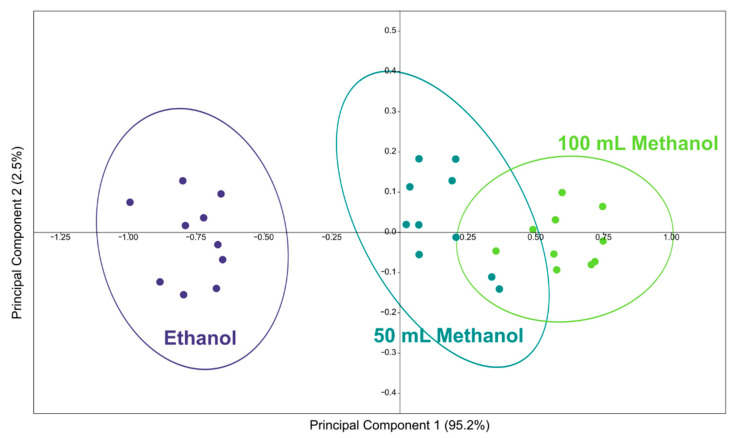
Principal components plot for alcoholic mixtures at 40° of pure ethanol in distilled water, with 50 mL of methanol, and 100 mL of methanol.

**Table 1 sensors-22-05554-t001:** Confusion matrix of the classification of the different types of measurements (type of gas and its concentration).

		Estimated					
		Ethanol			Methanol		
		1 ppm	10 ppm	100 ppm	1 ppm	10 ppm	100 ppm
True	1 ppm	8					
	10 ppm		8				
	100 ppm			8			
	1 ppm				8		
	10 ppm					8	
	100 ppm						8

## Data Availability

The data presented in this study are openly available in OSF with doi:10.17605/OSF.IO/MYXBH.

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
