# Peer review of "Nanosensor Based on Thermal Gradient and Machine Learning for the Detection of Methanol Adulteration in Alcoholic Beverages and Methanol Poisoning"

_sensors, 2022, doi:10.3390/s22155554_

Round 1

Reviewer 1 Report

The paper describes the detection of methanol in alcoholic beverages using a resistive nanosensor based on tin oxide nanowires with multivariate analysis and machine learning.  The study is interesting and the paper is written well.  The PCA and LDA demonstrated accurate separation between ethanol and methanol, while the SVM results clearly show high accuracy in predicting the concentration. The paper is recommended for publication after a few minor revisions and the authors may revise according to the comments below.

-       The authors mentioned that the differences in temperatures will somehow simulate different materials: “Chemical reactions depend on temperature and therefore, by combining the response at different temperatures, we obtain a series of different responses, as if they came from sensors made of different materials.”  I think this statement is not needed and not quite accurate as we know that although materials tend to perform differently at various temperatures, the performance is still dependent on the intrinsic properties of the material.  I recommend not including this statement.  The model in this study may not necessarily work on other materials and a new training/testing/tuning will have to be done.

-       The “realistic conditions” in section 3.7 only used a single concentration (proof), however, the solutes present in the actual alcoholic drink may affect the headspace concentration and have different actual responses for the same methanol concentrations. 

-       The principal component 1 (98.8%) clearly shows that it explains most of the variances while component 2 only accounts for a very small fraction (1.1%).  Can the authors explain which feature/s from the 5D dataset is/are the most important?

Reviewer 2 Report

The authors of “Nanosensor Based on Thermal Gradient and Machine Learning for the Detection of Methanol Adulteration in Alcoholic Beverages and Methanol Poisoning” report a new resistive gas sensor based on tin oxide nanowires. The work considers a wide temperature range and the presence of relative humidity up to 75%. Several machine learning algorithms are employed to enable the distinction between methanol and ethanol. The work developed is interesting and might lead to future application in day-to-day life.

Some minor issues/questions are summarized below:
1) How does the reported sensor selectivity towards ethanol/methanol compares to others already published?

2) How were the working temperatures chosen? Why are no values below 180 ºC?

3) Figure 1a can be reduced in size.

4) Please add the temperature values of the curves presented in Figures 2c and 2d, although different colors were used and the legend is presented in Figure 2a it can also be repeated in the other figures.

5) In Figure 3 use the same green and blue shades of ethanol and methanol points in the inset graph.

6) Did the authors expect such a high “mobility” of the points corresponding to 100 ppm of methanol with the increase in RH (Figure 5)? Why isn’t this “mobility” observed for the same concentration of ethanol?

7) I did not understand if mixtures of ethanol and methanol were considered in the subsection 3.7? Were the mixtures purely ethanol and methanol?

8) The references need to be uniformized, some present all the authors (5 or more) while others present two and et al.; Some journals names aren’t abbreviated; In general, when websites are used as references the authors should add the date of the website consultation.

9) Although it might be an Editorial question, please add spaces between the text/figure captions and the subsections; keep the captions next to the figure or table (same page).

Reviewer 3 Report

A resistive gas sensor based on tin oxide nanowires was proposed by the authors in this paper. The resistance change of the resistive sensor comes from the reaction of the volatile molecule methanol and ethanol with the probe, releasing or absorbing electrons in the process. And using machine learning algorithms (PCA, SVM, LDA) to analyze the data information collected in different temperature gradients to distinguish methanol and ethanol. This thermal gradient-based electronic nose is interesting. In order for articles to be published in journals of higher quality, the following issues need to be noted and properly explained:

1. Too many keywords, please reduce the number of keywords and make them more in line with the theme of the article.

2. The content corresponding to lines 96, 97, and 98 should not be used as a separate paragraph. It should be connected with the previous paragraph or explain what kind of performance is in this paragraph.

3. The specific value or range of "limit pressure" in line 108 should be mentioned.

4. How much does the concentration of nanowire forest mentioned in line 123 in the dimethylformamide dispersion affect the experimental results? Because I noticed that you only deposit a few drops of the dispersion onto the silicon wafer. Is it necessary to describe the relevant experimental data in detail?

5. Line 134 mentions the need for an 8-hour annealing treatment prior to testing to stabilize performance. Does this process have to be done once per inspection? This is not in line with the "rapid and inexpensive screening" mentioned in line 96, please check whether there is a problem with the language expression.

6. Is it reasonable that the chemical reactions mentioned in line 82 regarding the detection mechanism have not been characterized experimentally or specified in the article?

7. Whether the "the mean absolute error" mentioned in line 239 has positive and negative characteristics and direction, please check whether the expression is correct.

8. Please beautify the format of "10ppm" in Table 1.

Round 2

Reviewer 3 Report

Accept